

# A Fermi surface descriptor quantifying the correlations between anomalous Hall effect and Fermi surface geometry

Elena Derunova[1,2*], Jacob Gayles[3], Yan Sun[4], Michael W. Gaultois[5] and Mazhar N. Ali[2,6]

**1** Leibniz Institute for Solid State and Materials Research IFW Dresden, Germany
**2** Max Plank Institute of Microstructure Physics, Halle, Germany
**3** University of South Florida, Tampa, Florida, USA
**4** Max Planck Institute for Chemical Physics of Solids, Dresden, Germany
**5** Leverhulme Research Center for Functional Materials Design,
The Materials Innovation Factory, University of Liverpool, Liverpool, UK
**6** Delft University of Technology, Delft, Netherlands

⋆ derunova-el@mail.ru

## Abstract

In the last few decades, basic ideas of topology have completely transformed the prediction of quantum transport phenomena. Following this trend, we go deeper into the incorporation of modern mathematics into quantum material science focusing on geometry. Here we investigate the relation between the geometrical type of the Fermi surface and Anomalous and Spin Hall Effects. An index, $\mathbb{H}_F$, quantifying the hyperbolic geometry of the Fermi surface, shows a universal correlation ($R^2 = 0.97$) with the experimentally measured intrinsic anomalous Hall conductivity, of 16 different compounds spanning a wide variety of crystal, chemical, and electronic structure families, including those where topological methods give $R^2 = 0.52$. This raises a question about the predictive limits of topological physics and its transformation into a wider study of bandstructures' and Fermi surfaces' geometries and relating them to the quantum geometry theory of a more general metric of eigenstates, opening horizon for the prediction of phenomena beyond topological understanding.

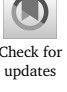

# 1    Introduction

Topological materials have garnered significant attention in recent years for their remarkable electronic properties, which have opened new horizons in condensed matter physics and electronic device engineering [1]. Among other effects, they exhibit anomalous Hall effect (AHE), i.e. additional orthogonal Hall current, which is unexpected from the classical electromagnetic theory perspective. There are various contributions to the AHE, but in this work, we only consider the intrinsic one, which is AHE generated by the internal magnetism combined with the properties of the electronic eigenstates $\psi_{nk}$ [2]. The accurate prediction of Anomalous Hall Conductivity (AHC), quantifying AHE in materials, especially within the realm of topological materials, relies upon sophisticated computational methodologies. These methods are rooted in the computation of Berry curvature $\Omega(\mathbf{k})$, a fundamental concept in condensed matter physics that offers insights into the geometric phase acquired by electron wavefunctions as they traverse the Brillouin zone. This enables accurate predictions of AHC by establishing a clear connection between the topological properties of electronic bands and transport phenomena as follows [2]:

$$\sigma_{xy} = -\frac{e^2}{\hbar} \int_{\mathrm{BZ}} \frac{d^2k}{(2\pi)^2} \Omega_{xy}(\mathbf{k}) f^{Fermi-Dirac}(\epsilon(\mathbf{k})). \tag{1}$$

Computational procedures to predict AHC typically commence with ab-initio density functional theory (DFT) calculations. These calculations are followed by a projection into Wannier functions, and subsequently, the computation of Berry curvature at each k-point in the Brillouin zone. The Kubo formula, which relies on Berry curvature and integrates over occupied states in k-space, is then employed to predict the AHC [3]:

$$\Omega^z_{xy,n}(k) = \sum_{m \neq n} Im\left[ \langle \psi_{nk}| v_x |\psi_{mk}\rangle \frac{\langle \psi_{mk}| v_y |\psi_{nk}\rangle}{(\varepsilon_{nk} - \varepsilon_{mk})^2} \right]. \tag{2}$$

While this approach has made significant strides in comprehending and forecasting the AHC in topological materials, challenges persist in, e.g. algorithmizing the computational procedure or attaining a high degree of accuracy. The accuracy issues appeared even with simple compounds: while for *Co* and *Fe* Berry curvature based predictions give reasonable errors within 30% compared to the experiment, the *Ni* predictions have about 250% mismatch [4]. There are various reasons for that, but one could be that the current computational method fundamentally does not take into account the possible coexistence of multiple *unconnected sets* of bands, having one elementary band representation (EBR) at the Fermi level, as was recently presented in topological quantum chemistry as multi-EBR bandstructures [5]. Another possibility is that there are additional contributions to the AHC beyond those accounted for by the Berry curvature. A more crucial factor for the numerical simulation of quantum effects like AHE is a pressing need to streamline the computational complexity of these calculations to enable high-throughput performance and online execution within materials databases.

A possible key avenue for achieving this computational streamlining can be found in consideration of the Fermi surface (FS), representing the locus of states with the highest occupied energy levels, which is fundamental in shaping electronic transport phenomena. However, a

significant challenge arises in the fact that the topological connectivity described by the Berry curvature is typically regarded as an intrinsic property of eigenstates and is not readily expressible solely through the distribution of eigenvalues. Nevertheless, an indirect link between the Fermi surface shape and the topological properties of the eigenstates can be assumed. Since the corresponding eigenvalues represent the action of the Hamiltonian on the eigenstates, i.e. $\varepsilon_n(k) = H(\psi_{nk})$, when H behaves, for instance, as a homeomorphism, it preserves the topological properties [6], i.e. topological connectivity remains for $\{\varepsilon_n(k)\}_{n,k}$ and thus it should be reflected in the shape of the FS. However, a detailed analysis of the exact expressions for the equivalents of the Berry connection and curvature on the space $\{\varepsilon_n(k)\}_{n,k}$ falls outside the scope of this work. Instead, we focus on a preliminary qualitative exploration of the FS shape in relation to anomalous quantum transport phenomena, utilizing the standard Riemannian metric on $\{\varepsilon_n(k)\}_k$.

In this context, our paper introduces a groundbreaking phenomenological Fermi surface descriptor $\mathbb{H}_F$ related to the density of hyperbolic points on the Fermi surface, whose computation does not include any topological quantities, but only Fermi surface data. However, it shows a higher match to the AHC compared to current methods and unlike them can be simply integrated into material databases. This empirically developed index correlates extremely well with experimentally measured values of intrinsic anomalous Hall conductivity (AHC) ($R^2 = 0.97$, whereas current methods gives just $R^2 = 0.52$). $\mathbb{H}_F$ is tested on 16 different real materials that broadly range from conventional ferromagnets to Weyl semimetals, including cases like Ni and $Co_2MnAl$, where the Berry phase approach (via the Kubo formalism) does not represent a complete picture of the transport. We found that 13 of the compounds have one EBR FSs and that the limit of the AHC for a single EBR FS is $\approx 1570\ (\Omega cm)^{-1}$. Two of the materials examined here, $CrPt_3$ and $Co_2MnAl$, have multi-EBR Fermi surfaces and subsequently break the apparent AHC limit. We also find that the $\mathbb{H}_F$ matches predictions of the spin Hall conductivity (SHC) for Pt, Beta-W ($W_3W$), and $TaGa_3$. The $\mathbb{H}_F$ index also enables an inexpensive and rapid computational prediction of AHE/SHE materials and can be implemented with existing density functional theory (DFT) methods and databases.

Our phenomenological research not only provides a valuable tool for practical applications in the prediction of AHC, but also underscores limitations in conventional transport theories and suggests possible directions for further theoretical exploration. In the following sections, we outline the development and application of the description $\mathbb{H}_F$, providing evidence of its predictive capabilities and highlighting its potential impact on the field of topological materials research and its technological applications.

## 2 Semiclassical dynamics and Fermi surface geometry

In our study, we rely on a semiclassical approach, which helps to understand how electrons move when subjected to electric ($\mathbf{E}$) and magnetic ($\mathbf{B}$) fields. Applied fields generate dynamics in the momentum space described by the following equation [7]:

$$\frac{d}{dt}(\hbar\mathbf{k}) = -e\left(\mathbf{E} + \mathbf{v} \times \mathbf{B}\right).\tag{3}$$

Since $\mathbf{v}$ in this equation is the electron's velocity defined by $\mathbf{v} = \frac{1}{\hbar}\nabla\varepsilon_n(k)$, which is a normal vector to the Fermi surface defined by $\varepsilon_n(k) = E_F$, the magnetic field generates dynamics on the Fermi surface that lead to the existence of so-called cyclotron orbits around the FS in the plane perpendicular to the applied field (see figure 1A, magnetic field in the x direction, orbits indicated by the purple lines). Thus, the shape of the FS must influence the dynamic in the magnetic field described by the equation 3 and, hence, it should affect transport properties.

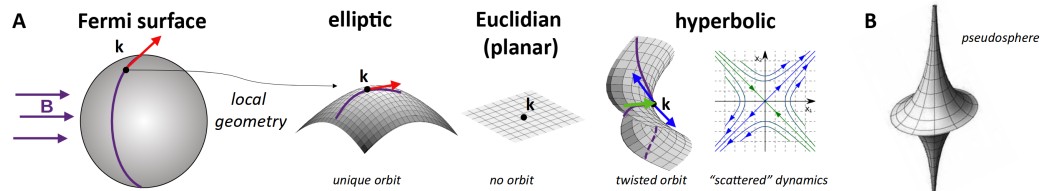

Figure 1: **(color online):**Local Fermi surface geometry and Fermi surface orbits in a magnetic field. A. Three possible local geometric types (left to right): elliptic, planar, and hyperbolic. For the hyperbolic case, the orbit's projection onto the local coordinate system $(x_1, x_2)$ around the point of splitting orbits is shown on the right. B. An example of a fully hyperbolic surface.

From that perspective the "shape" of the Fermi surface can be fundamentally classified only into 3 types: elliptic, planar, and hyperbolic, representing positive, zero, and negative Gaussian curvature of the surface correspondingly [8] (see figure 1). Cyclotron orbits are most commonly discussed in the context of convex closed FSs, which implies ellipticity from a geometric standpoint. However, the presence of hyperbolic points on the FS is known to affect transport properties in magnetic fields [9]. On the one hand, they may generate open orbits, leading to high magnetoresistance [7, 10, 11]. On the other hand, they break the convexity of the FS, which then also leads to non-trivial transport effects [12, 13]. Typically studies investigate the effect of a single hyperbolic point as in e.g. in the Lifschitz transition of convex-concave FS through the critical point [14, 15] or in the magnetic breakdown and quantum tunneling between orbits in topological semimetals [16]. Despite significant progress, capturing the impact of the whole hyperbolic regions of the Fermi surface — areas consisting entirely of hyperbolic points — on transport properties remains a big challenge, as we explain below.

To illustrate difficulties appearing with the hyperbolic surfaces, we consider closely equation 3, without an applied electric field, so it can be written as follows:

$$\frac{d}{dt}\mathbf{k} = -\frac{e}{\hbar^2}\xi, \qquad \xi = \nabla\varepsilon_n(k) \times \mathbf{B} \in T_k(FS). \tag{4}$$

These types of equations are known as dynamical systems and have been extensively studied. Equation 4 particularly, since the vector $\xi$ lies within the tangent space of the Fermi surface, $T_k(FS)$, by construction, describes a geodesic flow, meaning motion along the shortest path on the Fermi surface in the direction of $\xi$. For a sphere, geodesic lines are simple cyclotron orbits (figure 1A, left). However, when a hyperbolic point is present, orbits become more complex and self-intersecting. In this case, one direction is stable, where the orbit approaches the hyperbolic point, while the other is unstable, where it diverges, making both escape directions equally probable and introducing uncertainty in the dynamics. If the vector $\xi$ aligns with the stable direction, the dynamics deviate from the expected cyclotron motion and momentum scatter along the unstable direction (blue arrow in figure 1A, right) The formation of orbits and dynamics near hyperbolic points have been thoroughly studied by A. Y. Maltsev and S. P. Novikov in e.g. [17].

Even individual momentum shifts at hyperbolic points can generate complex, chaotic orbits where momentum can scatter between orbits, as demonstrated in [18]. In the presence of hyperbolic regions on the FS, the mixing of electron orbits becomes increasingly intricate and unpredictable. In the extreme case of a fully hyperbolic Fermi surface, such as e.g. the pseudosphere shown in figure 1B, it has been found that the geodesic flow exhibits ergodic behavior [19]. This means that, over time, the orbits uniformly cover the entire hyperbolic surface, and the dynamics become fully chaotic, rendering any individual orbit indistinguishable. Such chaotic behavior presents a significant challenge to traditional transport theories.

However, it is important to stress that ergodic dynamics have only been rigorously established for systems exhibiting negative curvature all over the surface, a condition most likely not met by the Fermi surfaces of real materials. Therefore, there might be a possibility of adjusting conventional theories to include the effect of the hyperbolic dynamics. E.g. it is reasonable to *hypothesize* that regions of negative curvature on the Fermi surface give rise to stochastic dynamics, effectively acting as "black boxes" where momentum can unpredictably shift during semiclassical dynamics by the "diameter" $D_H$ of the hyperbolic region. While rigorously quantifying this effect on transport remains challenging, we can formulate simple *qualitative phenomenological principles* that influence its magnitude.

- The more hyperbolic regions, the stronger the effects of the stochastic momentum dynamics on transport.

- The more elliptic regions in between hyperbolic regions, the more stochastic dynamics dissipate back to the semiclassical, and consequently contribute less to transport.

We emphasize that the same reasoning can be applied not only to the FS, but also to the energy band and the dynamics governed by $\frac{dx}{dt} = \frac{1}{\hbar}\nabla\varepsilon_n(k)$. Consequently, non-trivial transport in insulating and semiconducting materials can also be linked to the underlying geometry and hyperbolic dynamics. In this case, however, since the band is a 3-dimensional manifold, there are not only more geometric types, but the dynamics on such manifolds is an active area of study. For example, the entropy formula for certain dynamical systems on them was discovered only recently, which famously led to the proof of the Poincaré conjecture [20]. For now, we aim to conduct a *preliminary qualitative numerical study to provide compelling evidence that justifies a more rigorous investigation into the geometric description of transport theories*. From that perspective, one of the most studied quantum effects is AHE, for which we can make a comparison with the experimental results. We assume that described above hyperbolic scattering of the semiclassical dynamics can result in an anomalous Hall current, similar to how it appears under non-adiabatic evolution of eigenstates due to the Berry curvature, hypothesizing that the presence of hyperbolic areas on the FS is implicitly governed by topological connectivity. Building on the two phenomenological principles discussed earlier, we construct an empirical measure to quantify the effect of hyperbolic dynamics on the Fermi surface.

## 3 Results

### 3.1 Hyperbolicity index $\mathbb{H}_F$ and anomalous Hall conductivity (AHC)

To more rigorously quantify the correlation between hyperbolic regions on the FS and AHE/SHE in a chosen direction we introduce the index of the concentration of the hyperbolic areas of the FS, which we denote by $\mathbb{H}_F$ and define as the following:

$$\mathbb{H}_F = \frac{S_{hyp}^\alpha}{S_{tot}^\alpha}. \tag{5}$$

Where $S_{hyp}^\alpha = \sum_{FS} I_n|\sin\alpha|\Delta k^2$, $S_{tot}^\alpha = \sum_{FS}|\sin\alpha|\Delta k^2$, $\alpha$, is an angle between the tangent plane to the FS and AHE plane, and $I_n$ is the sign of the $\partial^2\epsilon_n/\partial^2 k$ at the BZ boundary in the Hall current direction. The angle between the tangent plane and the AHE plane indicates effect on the direction of AHE, considering that AHE is maximum when the splitting of orbits happens in the orthogonal to the applied field plane. As defined, $\mathbb{H}_F$ can have a maximum of "1" which means that the entire FS would be hyperbolic, except for the points where the tangent

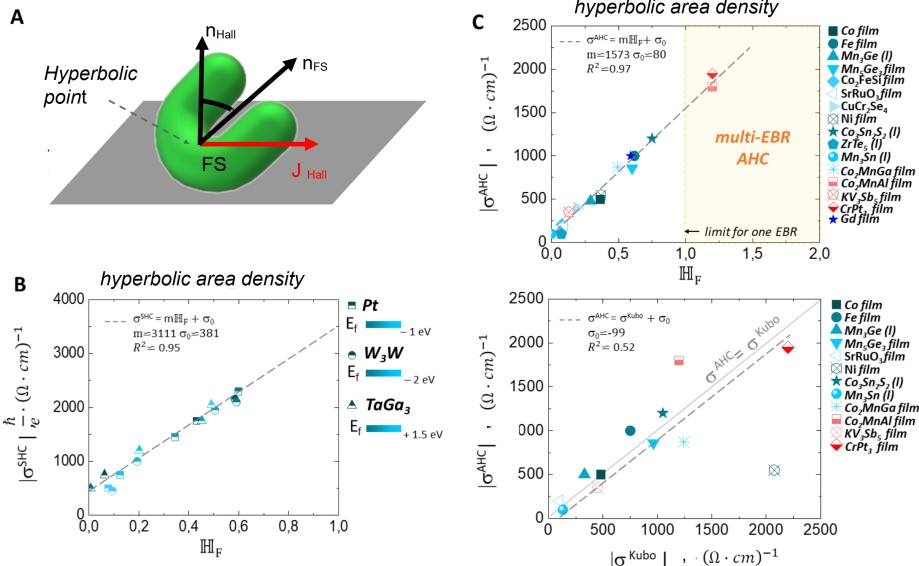

Figure 2: **(color online):** A. Schematic image of the $\mathbb{H}_F$ calculation in the direction of Hall measurement. B. Correlation graph of the predicted SHC values via the Kubo formalism vs $\mathbb{H}_F$, as defined in the text. C. Correlation graph top: *experimentally* determined intrinsic AHC vs $\mathbb{H}_F$ for various 2D or layered materials ((l) identifies layered structures). Bottom: *experimentally* determined intrinsic AHC vs predicted values of AHC via the Kubo formalism.

plane is orthogonal to the AHE plane. Assuming that the shape of the FS reflects topological connectivity, the summation is performed over all connected bands (i.e., one EBR bands) that form the FS. This approach implicitly accounts for inter-band exchange by positing that the dynamics governed by the semiclassical equation 3 occur across different bands within one EBR.

We performed unperturbed DFT calculations of 16 compounds for which intrinsic AHC values were rigorously experimentally determined [21–29], covering a variety of structural families (perovskites, Heuslars, kagome lattices, FCC lattices, etc.) and topological classes (Dirac/Weyl/Trivial metals and semimetals). Quantum anomalous Hall insulators and quantum spin Hall insulators, however, cannot be included in our consideration as they don't have a Fermi surface and the influence of geometry needs to be considered differently. We compared those experimental AHC values (mainly from [21]: its supplementary information table S3) to our calculated $\mathbb{H}_F$ (taking care to align the directions of calculation with the directions of measurement for each material in the various experiments). Since the $\mathbb{H}_F$ parameter is currently defined for 2D conductance, where transport effects in the third dimension are negligibly small, we calculated $\mathbb{H}_F$ for thin films or layered crystal structures (indicated as (l) in Figure 2), where the layers have enough separation, so the contribution of the third dimension to the overall effect is relatively weak. In these cases, the AHE can reasonably be considered quasi-2D and compared to the $\mathbb{H}_F$.

The result, shown in figure 2, shows an extraordinary linear correlation of the concentration of hyperbolic areas of the FS with the experimentally measured AHC values of all compounds, regardless of structural family or topological class, with an $R^2$ value of 0.97 ( figure 2c). Even though as defined $\mathbb{H}_F$ does not give any predicted value of AHC, the found slope value of $m = 1573$ of the correlation can used as an empirical normalizing factor for the use of $\mathbb{H}_F$ as a predictive descriptor of AHC so that $\sigma^{AHC} = w\mathbb{H}_F, w = 1573(\Omega cm)^{-1}$. For compar-

ison, we also present a plot of the experimentally measured intrinsic AHC values versus the calculated AHC values using the Kubo formalism (based on Berry curvature [4,21,24–28]) for the same compounds (Figure 2d). The $R^2$ drops down to only 0.52, with a few exceptionally inaccurate cases like $Co_2MnAl$ or Ni, where the error is large and the reason is still under investigation [30]. Even without taking those two compounds into account, the $R^2$ from the Kubo calculated AHCs only rises to 0.87; significantly worse than the $\mathbb{H}_F$-dependence.

We made a similar calculation for SHE compounds, but due to a paucity of experimental data, we are forced to plot the comparison of the Kubo predicted SHC values for Pt, $W_3W$ [3,31] and $TaGa_3$ (See figure 6 in supplement) at different $E_F$-levels against their corresponding $\mathbb{H}_F$ values in Figure 2b. This graph also shows a strong correlation of the concentration of hyperbolic areas with the Kubo calculated SHCs with an $R^2$ of about 0.95. Such an extraordinary correlation with Berry curvature-based method would be highly unlikely if $\mathbb{H}_F$ was entirely unrelated to the topological quantities. This numerical evidence suggests a potential for expressing transport dominated by the topology of eigenstates in terms of the band's geometry, supporting the hypothesis of a deeper connection between the two. Exploring their relationship more rigorously could be a promising direction for future research. Interesting questions would include finding theoretical expressions of the slope value in both graphs (Figure 2b and 2c, top) and the y-axis offset in Figure 2b, as they may have fundamental meaning.

## 3.2 Correlation-based prediction of AHC: Multiple-EBR Fermi surface

From the correlation in figure 2, it can be seen that in the limit of $\mathbb{H}_F = 1$, the intrinsic AHC is expected to reach a maximum value of 1570 $(\Omega cm)^{-1}$ according to the slope of the linear fit. However, there are two compounds ($Co_2MnAl$ and $CrPt_3$) that have $\mathbb{H}_F$ and intrinsic AHCs greater than these maxima. While at first this appears to be an inconsistency, the limit on $\mathbb{H}_F$ can be broken if we take into account the EBR (elementary band representation) for the bands forming FS. Recently it was shown that all bands can be grouped into sets that correspond to distinct EBRs; topological semimetal behavior can be understood as a property of a partially occupied set of such bands. Also, the non-quantized contribution to the AHE, as shown by Haldane et al [32], is expected to be a pure Fermi surface property. Combining these two ideas, a part of the FS that is comprised of multiple pockets created by the bands belonging to a single EBR, can be considered distinctly from *another part* of the FS similarly corresponding to the *bands from another EBR*.

In the common case where there is a continuous gap disconnecting sets of bands contributing to the FS, $\mathbb{H}_F$ can be calculated separately using formula 5 for parts of the surfaces arising from distinct sets of bands (bands with differing EBRs), essentially dividing the bandstructure into its connected components by their EBR, and subsequently summed together in order to characterize the entire FS. This is exactly the case for $Co_2MnAl$, $CrPt_3$ and $KV_3Sb_5$. For the case of $KV_3Sb_5$, it can be seen that the contributions of the distinct EBRs are not cooperative, resulting in a relatively low $\mathbb{H}_F$ of 0.14. However, for $Co_2MnAl$ and $CrPt_3$, both have cooperative contributions and correspondingly have $\mathbb{H}_F$ values larger than 1 as well as AHC values larger than 1570 $(\Omega cm)^{-1}$; but they still correlate extremely well with the overall trend in Figure 2c.

Figure 3a showcases the detailed bandstructure for $CrPt_3$ with each distinct set of bands colored (blue and yellow) with the continuous gap shaded in gray. The insets clarify the almost-degeneracies near Gamma which are actually gapped. In the Berry curvature approach, the states from the different EBRs are mixed in the total calculation in the Kubo formula. figure 3b shows the energy dependent AHC calculated from the Kubo formalism as well as the energy dependent AHC (using the AHC vs $\mathbb{H}_F$ correlation $\sigma = m\mathbb{H}_F + \sigma_0$ to convert $\mathbb{H}_F$ to a numerical AHC value). The results from the two methods are qualitatively similar, but the $\mathbb{H}_F$ result has a slightly better quantitative match to experiment.

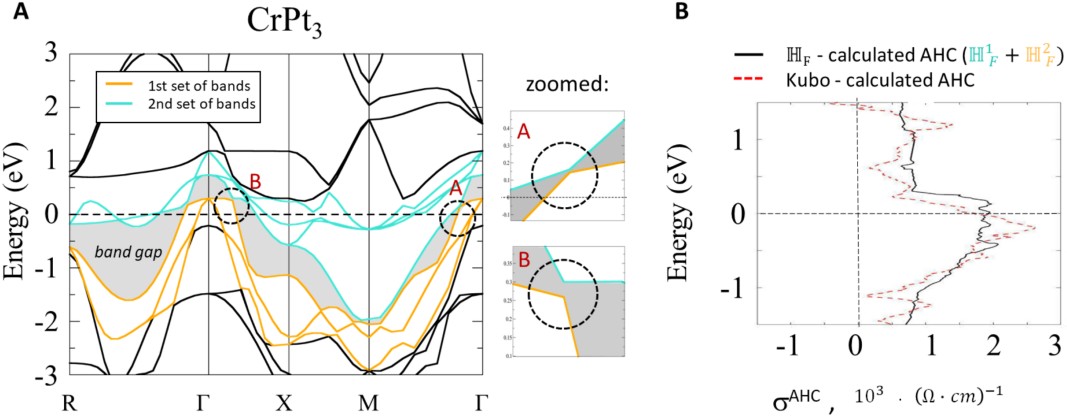

Figure 3: **(color online):** A. Bandstrucure of $CrPt_3$. Blue and yellow colors represent two topologically disconnected (having different EBRs) sets of bands crossing the Fermi level. These sets are disconnected by the continuous gap present between them; i.e. true semimetallic behavior. B. Graph of energy resolved AHC predicted in two different ways: red dashed line is the Kubo based prediction, black dashed line stems from the linear correlation between $H_F$ and AHC calculated separately for FS contributions from each set of bands, then summed together for total $\mathbb{H}_F$.

## 4 Discussion and conclusion

Why does the $\mathbb{H}_F$ method appear to fare better than the Kubo approach for these materials and properties? This is a wide area of future investigation, however, there are few important considerations we elaborate on here. Firstly, the $\mathbb{H}_F$ has a different theoretical motivation and can capture something beyond Berry curvature method, e.g. related to the quantum geometry effects [33]. Besides that, the Kubo formalism looks at the Berry curvature in a point-wise fashion without consideration of their connections to each other, and incorporates a fictitious broadening parameter that does not fully capture finite temperature effects to the electronic structure. Looking at independent points in momentum space means that if a particular point of importance is missed (because of, for example, a very sharp feature or a too low resolution $k$-mesh grid), its entire contribution is missed and the calculation can become inaccurate. This is fundamentally different than the *path-wise* $\mathbb{H}_F$ method which looks at points and their connections to each other because it is approximating trajectories. This is likely related to the $\mathbb{H}_F$ plateaus (see figure 4) at relatively sparse K-meshes of $\approx$30 x 30 x 30, unlike the typical >150 x 150 x 150 $k$-mesh grids used in the Kubo analyses (where the $k$-mesh must also increase for tightening the broadening factor). Secondly, the $\mathbb{H}_F$ calculations rely purely on the first principles calculation of the Fermi Surface and not an interpolated/tight-binding representation of the first principles calculation as in Kubo. This (i) eliminates the need for Wannier/tight-binding Hamiltonians which loses the gauge invariance in the convergence process, and (ii) $\mathbb{H}_F$ calculations are not restrained by the quality of the Wannier fit which are localized functions that will always have trouble capturing the completely de-localized topological states present in some systems. Finally, and perhaps most importantly, the Kubo formalism looks at two-band intersections, not multiband intersections, meaning it ignores higher order intersections that can also result in anomalous transport contribution. The $\mathbb{H}_F$ method inherently looks at *n-band intersections* since it is a pure Fermi Surface analysis method and the Fermi Surface (and its features) is made up of any number of bands crossing $E_F$.

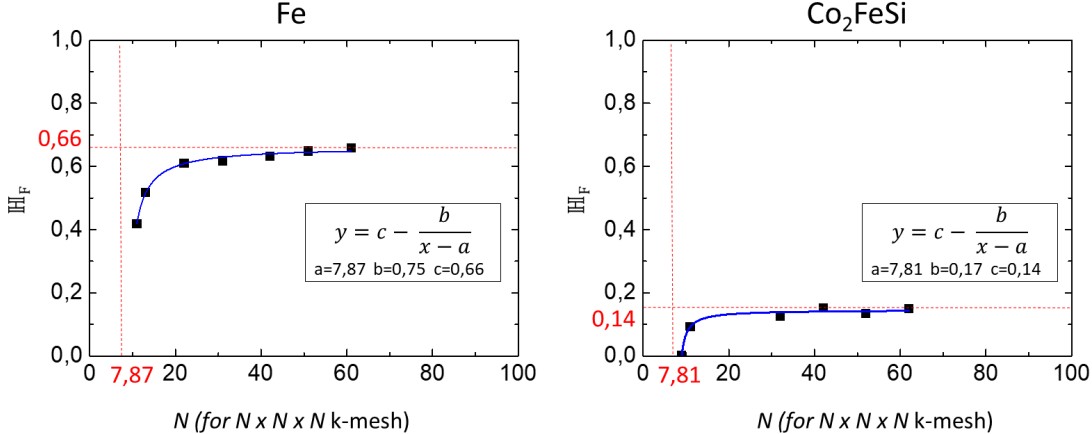

Figure 4: **(color online):** $\mathbb{H}_F$ dependence of k-mesh density for Fe and $Co_2FeSi$. y-axis is the calculated total $\mathbb{H}_F$, and the x-axis is k-points cubed.

When compared with the current Berry curvature driven method for AHE/SHE prediction via the Kubo formalism, the $\mathbb{H}_F$ index is also, computationally, a much simpler metric as it requires just basic DFT calculation without Wannier projection, and thus can carried out at a significant reduction in time and cost. Importantly, this analysis method can easily be fully automated and implemented into material databases, and can enable artificial intelligence and machine learning based searches of large repositories of compounds for materials with desirable traits for technological applications. For now the $\mathbb{H}_F$ index is still a relatively rough estimation and also is limited to the cases of quasi 2D materials. However, the numerical correlation of the AHE/SHE with $\mathbb{H}_F$ of $R^2 = 0.97$ proves that the concept of using geometric classification is not just a "blue-sky" theoretical research effort; it has immediate applications to outstanding questions in condensed matter physics.

Important future work includes exploring the full theoretical motivation of the method and its relation to the topological transport theory. The graphs in Figure 2b and Figure 3b suggest that hyperbolicity may result from the topological connectivity of the eigenstates. However, a direct equivalence with the Berry curvature cannot be concluded. For instance, as seen in Figure 3b, $\mathbb{H}_F$ follows the general trend of Berry curvature-based values but smooths and averages its peaks. This may imply that $\mathbb{H}_F$ reflects not only the Berry curvature but also a more general metric that encompasses it, such as e.g. the Fubini-Study metric related to the quantum geometry effects [33]. The Fermi surface (FS) analysis of Ni provides implicit confirmation of this. Figure 5 shows the distribution of hyperbolic points on the FS, classified into two distinct sets: smooth points (blue) and singular points (red), representing regions with low and high mean curvature, respectively. This classification reveals the crucial role of different hyperbolic point origins in the anomalous Hall effect (AHE). The singular points (red) most likely originate from Dirac points, where the linear dispersion of the type II Dirac cones creates sharp, cone-like cross-sections of the Fermi surface (see figure 7). These conical intersections generate regions of high mean curvature because the Fermi surface must rapidly bend around the singular geometry of the cone: near the Dirac point vertex, the surface transitions abruptly from one cone face to another, creating localized regions where the principal curvatures are large. This abrupt change in surface orientation around the Dirac cone vertices results in high mean curvature, while formally remaining within the hyperbolic geometric class. If only these Dirac-related contributions were considered, they would produce uncompensated contributions from the valence band, leading to high total anomalous Hall conductivity (AHC). However, the smooth hyperbolic points (blue) arise purely from the intrinsic hyperbolic geometry of the Fermi surface, unrelated to Dirac points, and exhibit lower mean curvature due

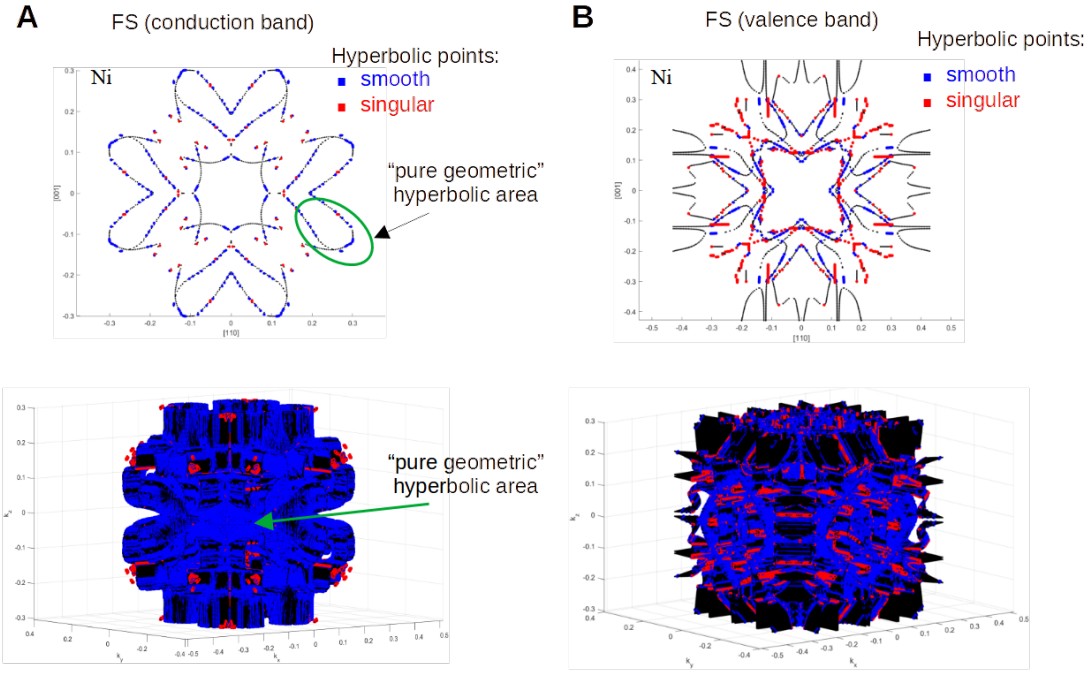

Figure 5: **(color online):** Fermi surface of Ni: A. conduction band, B. valence band. Blue represents hyperbolic points with low mean curvature (smooth), and red represents hyperbolic points with giant mean curvature (singular).

to the smoothness of the band dispersion around it. When both populations are included in the AHC calculation, the blue points provide compensation that significantly reduces the total AHC, bringing the theoretical predictions into agreement with experimental observations. While this mean curvature-based separation of hyperbolic points represents a non-rigorous estimate, it nonetheless provides reasonable insights into the origins of intrinsic AHC beyond the conventional Berry curvature and Dirac point contributions. The analysis illustrates that "pure geometric" hyperbolic points on the Fermi surface effectively facilitate tunneling between different FS regions, analogous to the inter-band exchange via Dirac points, thereby contributing to the AHE.

This geometric duality suggests a natural connection to the quantum geometric tensor (QGT) framework [33]. We propose that the QGT's complementary aspects may reflect this distinction: its imaginary part (Berry curvature) could encode momentum-space analog to "magnetic field" arising from band singularities like the Dirac points, representing the rotational aspects of the QGT, while its real part - related to the geodesic quantum distance on the band - might capture the intrinsic 3D band geometry and represent momentum-space analog to "electric field" related to the divergence properties of the QGT. For 3D energy bands, either the real part or the full QGT could potentially relate to the Thurston geometric classification of 3-manifolds, encoding the full geometrical "shape" of the band. We speculate that the hyperbolic features observed on the 2D Fermi surface - both the singular red points and smooth blue regions - emerge as cross-sectional manifestations of this underlying 3D band geometry. This quantum geometric perspective suggests that the compensating AHE contributions could arise from the interplay between topological singularities and the way the 3D band's geometry shapes the Fermi surface cross-sections.

In summary, we have introduced the idea of non-ergodicity of the FS orbits on the hyperbolic regions, which might result in the non-adiabatic evolution of the eigenstates and corresponding transport effects. This concept has been applied to develop a simple index, $\mathbb{H}_F$, for quantifying the contribution of the concentration of hyperbolic areas, and showed a universal correlation ($R^2 = 0.97$) with experimentally measured intrinsic AHE values for 16 different compounds spanning a wide variety of crystal, chemical, and electronic structure families. An apparent maximum value, at $\mathbb{H}_F = 1$, of 1570 $(\Omega cm)^{-1}$ was determined for materials with an FS created by bands belonging to a single EBR; materials with multi-EBR FS's can, and do, break this limit as evidenced by $CrPt_3$ and $Co_2MnAl$. Use of the $\mathbb{H}_F$ index allows direct calculation of the AHE/SHE at a much lower computational cost than current methods by eliminating the need for Wannier projection and can be implemented with existing high throughput DFT methods and databases. This work highlights the importance of, and opportunities laying ahead for, developing a complete theory of *geometrical understanding* of electronic structure manifolds beginning with Fermi surfaces. Also, these ideas can be extended to bosonic (e.g. magnonic) band structures and their constant energy momentum surfaces as well. In analogy to the broad impact that topological understanding of these structures had, a geometry incorporated in it may lead to a deeper understanding of at least electron transport and possibly have far-reaching consequences in condensed matter physics.

## Acknowledgments

**Funding information**    This research was supported by the Alexander von Humboldt Foundation Sofia Kovalevskaja Award and the BMBF MINERVA ARCHES Award. E.D. thanks O. Janson and Y. Blanter for the productive discussions and support on this work. M.W.G. thanks the Leverhulme Trust for funding via the Leverhulme Research Centre for Functional Materials Design.

**Author contributions**    E.D. was the lead researcher on this project. She carried out the main theoretical derivations as well as the majority of the calculations and analysis. Y.S., M.G. and J.G. assisted with calculations and interpretation. M.N.A and E.D. are the principal investigators.

# A Suplementary information

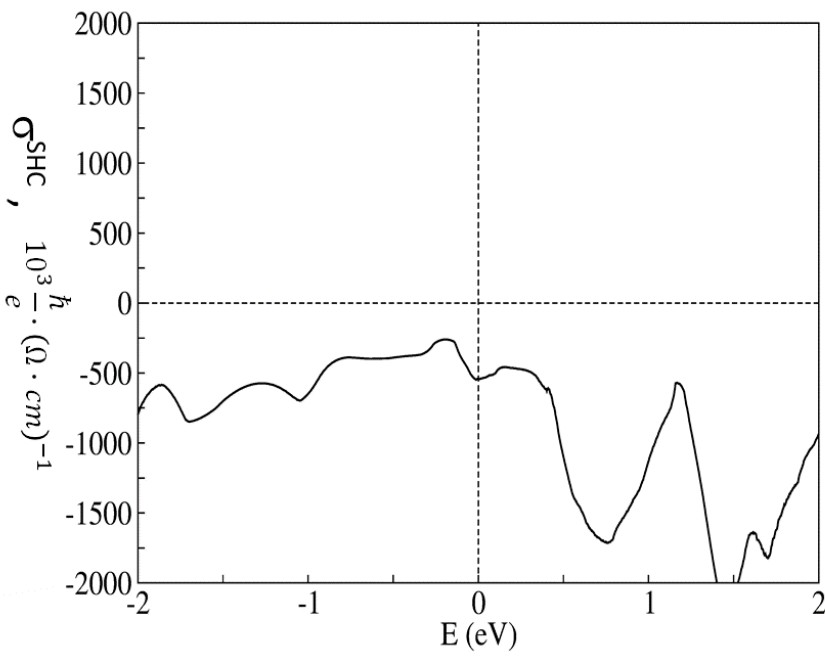

Figure 6: **(color online):** SHC vs Energy for $TaGa_3$.

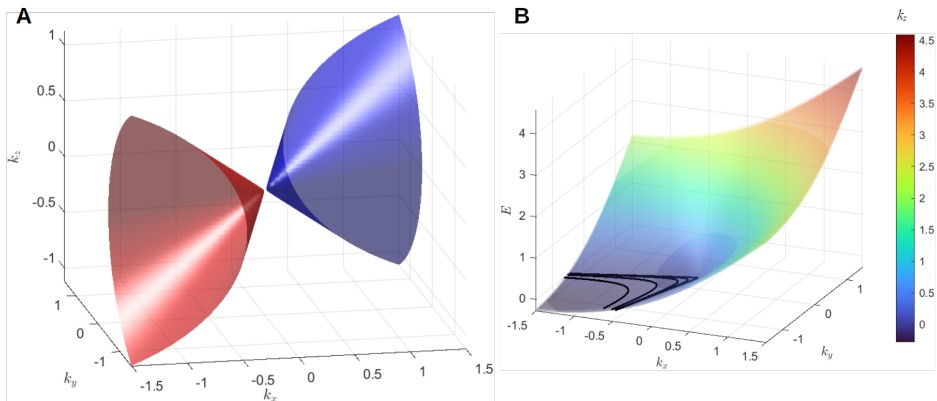

Figure 7: **(color online):** Fermi surface (A) and bandstructure (B) for a tilted Dirac cone. Black lines represent $E = E_f$ for different $k_z$, showing increased local curvature of the FS around the Dirac point.

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
