# Peer review of "A Fermi Surface Descriptor Quantifying the Correlations between Anomalous Hall Effect and Fermi Surface Geometry"

_SciPost Physics, doi:SciPost Phys. Core 8, 085 (2025)_

## Round 3 · Referee Report · Anonymous (Referee 1) · 2024-7-31

Strengths

An original idea is put forward and its confrontation with experimental data appears to be better than what is generally accepted regarding the amplitude of anomalous Hall conductivity in various materials.

The idea that the shape of the Fermi surface can lead to a non-adiabatic process, similar to what is caused by the Berry curvature, is intuitively appealing.

Weaknesses

I think the comparison of the new theory with the experimental data is not sufficiently rigorous and can be improved.

Report

The paper reports on an investigation of a possible relation between the geometry of the Fermi surface and the amplitude of Anomalous Hall Effect. The idea put forward is, to this reviewer’s knowledge, genuinely new. The authors define an index, which quantifies the hyperbolicity of the Fermi surface and then show that its amplitude correlates with the experimentally measured intrinsic anomalous Hall conductivity (AHC), of a variety of magnets. Remarkably, the correlation is stronger than the one between the measured and calculated AHC, based on the Berry curvature.

The paper is original, the topic is timely and I think the paper is important enough to be considered for publication in Scipost. However, I invite the authors to address the following issues:

i) My main criticism is that the comparison between what can be achieved by this approached and what was previously accomplished based on calculating the Berry spectrum does not look entirely fair. The upper panel of Figure 2c is a plot of the experimentally measured AHC vs the dimensionless hyperpolic factor. The lower panel is a plot is a plot of the experimentally measured AHC vs the theoretically predicted AHC. The authors argue that the correlation seen in the upper panel is stronger. This is convincing. But the completion is not fair. The upper plot does not make any prediction on the absolute value. In order to make their case more crystal clear, the authors should make a statement the slope of the upper panel in addition to the difference in the standard deviation in the linear fits. Moreover, what is the physical meaning of the finite intercept in the upper panel?
ii) For the same reason, I recommend that they include a table listing the experimental quantities and the theoretically expected values in the two competing pictures.
iii) As for experimental data, I strongly recommend to the authors to add more references and try to make an exhaustive list in order to dissipate any suspicion of cherry picking. It looks like that Dresden has been favored as a source of experimental data. Since in most cases, there is an experimental consensus, adding more references should strengthen and not weaken their case. Moreover, data on crystals are more reliable than data on films. I recommend to use the latter only when the former are absent, which is not the case of, for example Co2MnGa.
iv) I also think that the authors should discuss a number of specific cases. For example, what insight does this new approach provide in the specific case of nickel? What about KV3Sb5? The non-linear Hall response there arises in absence of magnetic order. Is it really an AHC or is it a feature of the geometry of the Fermi surface?
v) Finally, there are a number of minor issues:

• Page 2: “The accuracy issues appeared e.g. with simple compounds like Ni [4], where prediction gives a significant offset from experimentally observed AHC.” It may be a good idea to be more quantitative and specify that the agreement between theory and experiment is excellent for Co (477 vs. 480), not bad for iron (750 vs. 1032), and very bad for Ni (-2275 vs. -646).
• Page 2, introduction. It is worth to inform early the the uninitiated reader that EBR is “an elementary band representation”.
• Page 3 introduction. “Our research represents a significant advancement in the field of topological materials, offering a valuable tool for both theoretical investigations and practical applications.” I agree, but wouldn’t be wiser to avoid self-congratulation and let other investigators make such an acknowledgement?

Requested changes

See the report.

Recommendation

Publish (easily meets expectations and criteria for this Journal; among top 50%)

---

## Round 3 · Referee Report · Anonymous (Referee 2) · 2024-8-18

Report

The manuscript by Derunova et al. proposes HF, a hyperbolic Fermi surface index as a predictor of anomalous/spin Hall conductivity in magnetic and strong spin-orbit coupled materials. At the empirical level the discussions are valuable, and the approach adopted by the authors appears to reasonably reproduce the more well established Berry curvature scheme as shown in the case study of CrPt3, as well as the experimental anomalous Hall conductivities of quite a few systems.

However, I have identified a few major scientific concerns with the manuscript:
1. A general tone I sensed from the manuscript is that it claims their approach is superior to the more conventional Berry curvature approach. However, after carefully reading the text, it will only be fair for the authors to state more clearly that the method in the manuscript is *empirical*. The derivation in section 2 does not establish that these two approaches are equivalent or directly related.

2. In line with the above point, it is important that the authors clarify the limitations of the hyperbolic Fermi surface approach. For instance, the HF index does not describe the anomalous Hall and spin Hall conductivities of quantum anomalous hall insulators and quantum spin Hall insulators (as there are no Fermi surfaces in these cases). The manuscript should explicitly address these limitations.

3. In the manuscript AHC is expressed in the format number*\hbar/e (Ohm cm)-1 a few times. The inclusion of \hbar/e is redundant and not correct.

4. The discussion in section 2 on the mixing of orbits in a hyperbolic Fermi surface is not clear. More clarification and literature references are needed. What is the direction of magnetic field in Fig.1 for the hyperbolic Fermi surface?

Other more minor points:

1. Introduction Paragraph 1: “The are various contributions to AHE…” should be “There are…”

2. Introduction Paragraph 4: “which computation does not include…” should be “whose…”

3. Results Paragraph 4: what is “W3W”? The same typo appears in Fig. 2B.

4. Co, Fe, Mn3Ge, CuCr2Se4 are strictly speaking not 2D materials. The statement that the approach used in the manuscript is limited to the cases of 2D materials is not accurate.

5. No supplementary materials are available albeit referenced a few times in the manuscript

Recommendation

Ask for major revision

---

## Round 4 · Referee Report · Anonymous (Referee 1) · 2025-9-8

Strengths

Original

Weaknesses

Little attempt to explore the significance of the observation

Report

The main result of this paper its its figure 2. It is indeed striking to see a correlation between the experimentally measured anomalous Hall conductivity and a parameter quantifying the hyperbolic geometry of the Fermi surface.
What disappoints me is the fact that the authors did not try to make the physical picture behind this interesting observation more transparent. The vertical axis corresonds to a measured quantity and has physical units. The horizontal axis is calculated and dimensionless. Therefore the slope has a dimension (siemens per cm) . Why is it of the order of quantum of conductance per average lattice parameter ?

The slope cannot be called "empirical" because one of the axes is COMPUTED not MEASURED . Contrast this with Kadowaki-Woods plot, for example.

I recommend immediate acceptance of this paper by Scipost Core.

Recommendation

Accept in alternative Journal (see Report)

---

## Round 4 · Author Response

The requested changes regarding insights on Ni were made.

---

## Round 4 · List of Changes

Added discussion on Ni in the Discussion and Conclusion section.

---

## Editorial Decision

published